# Development of a Precipitation-Area Curve for Warning Criteria of Short-Duration Flash Flood

Deg-Hyo Bae[1], Moon-Hwan Lee[1], Sung-Keun Moon[2]

[1]Department of Civil and Environmental Engineering, Sejong University, Seoul, 05006, South Korea
[2]Water Resources Research Division, Water Resources & Environment Research Group, Rural Research Institute, Korea Rural Community Corporation, Ansan-si, Gyeonggi-do, 15634, South Korea

*Correspondence to*: Deg-Hyo Bae (dhbae@sejong.ac.kr)

**Abstract.** This paper presents quantitative criteria for flash flood warning that can be used to rapidly assess flash flood

occurrence based on only rainfall estimates. This study was conducted for 200 small mountainous sub-catchments of the Han River basin in South Korea because South Korea has recently suffered many flash flood events. The quantitative criteria is calculated based on Flash Flood Guidance (FFG) which was defined as the depth of rainfall of a given duration required to cause frequent flooding (1~2 years return period) at the outlet of a small stream basin and was estimated using threshold runoff (TR) and antecedent soil moisture conditions in all the sub-basins. The soil moisture conditions were estimated during

the flooding season, i.e., July, August and September, over 7 years (2002~2009) using the Sejong University Rainfall Runoff (SURR) model. A ROC (receiver operating characteristics) analysis was used to obtain optimum rainfall values and a generalized precipitation-area curve (P-A curve) was developed for flash flood warning thresholds. The threshold function was derived as P-A curve because the precipitation threshold with a short duration is more related to basin area than any other variables. For a brief description of the P-A curve, generalized thresholds for flash flood warnings can be suggested for

rainfall rates of 42, 32 and 20 mm/h in sub-basins with areas of 22~40 km², 40~100 km² and >100 km², respectively. The proposed P-A curve was validated based on observed flash flood events in different sub-basins. Flash flood occurrences were captured for 9 out of 12 events. This result can be used instead of FFG to identify brief flash flood (less than 1-hour), and it can provide warning information to decision makers or citizens that is relatively simple, clear, and immediate.

## 1 Introduction

Flash floods are among the deadliest natural disasters, with significant socioeconomic effects and the highest average mortality rate among types of floods (Jonkman, 2005). Flash floods are generally associated with localized, intense rainfall events in small and medium watersheds. It is difficult to monitor and forecast flash floods due to the unusually short response time for these natural disasters. Additionally, climate change likely increased the number of extreme rainfall events and the risk of flash floods (Gregory and Mitchell, 1995; Palmer and Raisanen, 2002). Therefore, reliable flash flood

forecasting methods are necessary for flash flood response.

To judge flash flood occurrence, there are three methods: flash flood susceptibility assessment, the flow comparison method, and the rainfall comparison method (Hapuarachchi et al., 2011). Flash flood susceptibility assessment can be considered a useful first step in determining the contributing factors to the flash flood vulnerability (possibility of flash flood occurrence and degree of danger) of a catchment using limited data (Collier and Fox, 2003). The flow comparison method compares the model-driven flow value with the observed flooding threshold, which is a criterion for deciding whether flooding should be expected or not. However, this approach has some limitations for real-time flash flood forecasting because it requires long historical data and hydrological simulation to establish a flash flood modeling system. The rainfall comparison method compares threshold rainfall causing flooding flow with the forecast rainfall instead of comparing forecast with observed flows. This method is a tool to warn of an imminent flash flood and the typical method is FFG (Flash Flood Guidance) (Carpenter et al., 1999; Carpenter and Georgakakos, 1993). This method is commonly used for flash flood forecasting, as it is easily understood by the general public because it provides a qualitative criterion that can be used to intuitively determine whether a flash flood will occur.

Some recent studies suggest the limitations of FFG (Norbiato et al., 2008; Montesarchio et al., 2011; Gourley et al., 2012). The limitations of FFG are in the assumptions of spatially/temporally uniform rainfall and linear response, and the use of regional relationships to make inferences about ungauged locations. FFG performance in ungauged basins is less accurate (Norbiato et al., 2008). Recent studies tried to improve the warning accuracy. Schmidt et al. (2007) proposed a raster-based method to derive a gridded FFG (GFFG). Gourley et al. (2012) reported that FFG performs better than GFFG, but GFFG can detect spatial variability. Miao et al. (2016) established a strategy for flash flood warning that is based on the definition of rainfall threshold using distributed hydrological model. They claimed that physically based methodologies are more appropriate for flash flood forecasting. In South Korea, flash flood studies have also been performed. Bae and Kim (2007) provided the flash flood guidance using the Manning equation, GIUH (geomorphologic instantaneous unit hydrograph), and TOPMODEL (Beven et al., 1994). Lee et al. (2016) generated a gridded flash flood index using the gridded hydrologic components of the TOPLATS land surface model and a statistical flash flood index model. Recent studies have focused on the accuracy and spatial distribution of FFG.

However, South Korea has recently suffered many flash flood events in the mountainous regions. More than 64% of South Korea is mountainous and prone to flash floods with very short rainfall durations. Recent heavy rainfalls in South Korea have triggered flash floods and landslides that caused severe damage to infrastructure and resulted in dozens of deaths. Notably, the heavy rainfall events have resulted in several flash floods since 2000, such as events in 2005, 2006, 2008 and 2012 at several locations in South Korea. In particular, the hourly maximum rainfall exceeded 50 mm/hr in 2006 and 2011, most of the flash flood events in South Korea were caused by short rainfall duration of less than one hour. It is difficult to capture these flash flood cases using the methods presented in previous studies. Therefore, prompt flash flood warnings are necessary for citizens and decision-makers.

It is less important to estimate the soil moisture or runoff in the regions where flash floods occur frequently with short duration because the response time for a flash flood is limited. It is necessary to develop the criteria for intuitively judging

the likelihood of flash flood occurrence with short duration. Although FFG-based methods provide useful mechanisms for flash flood warning, the real-time estimates of soil moisture required in some regions are often challenging to acquire prior to rapid response against flash floods. In this study, we proposed quantitative criteria using a P-A curve for flash flood warning based on FFG due to the lack of observed flash flood events. Thus, a P-A curve was derived by using FFG, but we

validated the criteria by using observed flash flood events. Additionally, this study derives the importance of soil moisture estimation and which variable has the largest effect for deciding flash floods related to topography information. The proposed criteria and methodology will serve as an important tool for issuing flash flood warnings based on only rainfall information.

## 2 Study Area and Datasets

The study was conducted in small mountainous sub-catchments in the Han River basin. The Han River basin is located in the center of the Korean Peninsula at 36°30´~38°55´N and 126°24´~ 129°02´E. The watershed area spans over 26,356 km², or approximately 23% of the South Korean territory (Figure 1). The 660 sub-basins with areas of 0.1~ 179.8 km² were delineated using ArcGIS (as shown in Figure 2a). Figure 2b shows the relative frequencies of sub-basins with areas in different ranges. The average area of a sub-basin was 38.5 km², with a standard deviation of 25.7 km². Most of the sub-

basins were in the range of 20~40 km², with a relative frequency of approximately 40%. The reservoirs located in the Han River basin were identified and omitted from further analysis to remove the effect of surface runoff storage on threshold runoff. The reservoirs store surface runoff from the upstream area and reduce the contributing area for surface runoff at downstream locations. Among the 660 sub-basins, we selected head water basins and mountainous basins and removed artificial river basins. A total of 200 sub-basins were selected, as shown in Figure 3a. Figure 3b shows the relative

frequencies of the areas of the selected sub-basins. The average area of a selected basin was 43.1 km², with a standard deviation of 19.8 km².

Rainfall and soil moisture were the main datasets used to estimate Flash Flood Guidance. Rainfall data were obtained at 96 locations from the Ministry of Land, Infrastructure and Transport (MOLIT) and at 25 locations from the Korean Meteorological Administration (KMA). Rain gauges recorded data at 114 locations, and the resolution of each station was

about approximately 217 km$^2$ (approximately 15 × 15 km). The average annual precipitation was 1,390 mm, and the annual mean temperature was 11.5 ℃ over the 30 years of weather data from 1980 to 2009. More than 70% of the annual precipitation occurs during the flood season (June, July, August and September). The probability rainfalls for 1-hr at Seoul station are 52 mm/hr, 74 mm/hr, and 91 mm/hr for 3-year, 10-year, and 30-year return periods, respectively. A Digital Elevation Model (DEM) with a 30×30 m resolution and soil maps at a scale of 1:25,000 were obtained from the Water

Resources Management Information System (WAMIS) of South Korea. The soil moisture conditions were estimated using the SURR hydrologic rainfall-runoff model.

In addition to the observed weather and flow datasets, data were collected for actual flash flood events. The actual flash flood information was obtained from various sources, including print and electronic media, covering an 8-year period (2005–2012). Table 1 presents the locations, dates, times and maximum rainfall intensities of flash flood events in the Han River basin. Flash floods are common in the study area and occur almost every year. In 2011, several flash flood events occurred with different areas and dates.

## 3 Methods

### 3.1 QPC Computation

This study presents a method for deriving a P-A curve that represents the rainfall thresholds occurring during flash floods. The method is based on FFG analysis to avoid the need to estimate soil moisture conditions. Figure 4 presents the overall procedure used to evaluate the quantitative precipitation criteria (QPC) for flash flood warning. First, the mean areal precipitation and FFG were calculated by using topographic, meteorological data for the sub-basins in the study area. To obtain FFG at current time (t), which is a summation of threshold runoff (TR) and soil moisture deficit, threshold runoff at each sub-basin is estimated. The soil moisture conditions from actual rainfalls are simulated by using SURR model, and we can decide whether a flash flood occurred at certain basin by comparing this FFG value and that from 1-hr prior to the actual rainfall. In this experiment, it is assumed that if the observed MAP is larger than the FFG, a flash flood occurs.

ROC analysis is used to obtain the QPC for the flash flood warning, and a virtual rainfall (VR) of 1~100 mm/h with a 1 mm/h increment is used for comparison with observed rainfall (OR). The occurrence criteria for virtual flash floods (e.g., VR > FFG or VR < FFG) and the occurrence criteria for actual flash floods (e.g., OR > FFG or OR < FFG) are used to obtain ROC scores for rainfall rates of 1~100 mm/h in each sub-basin, as presented in Table 1. The virtual rainfall values that produce the maximum ROC score are selected in each sub-basin. Finally, a generalized precipitation–area curve (P-A curve) is obtained using selected rainfall rates that produce maximum ROC scores as a function of the relevant area of each basin. For a detailed description of Threshold runoff, FFG, SURR, and the estimation of the ROC score refer to sections 3.2 and 3.3.

### 3.2 Flash Flood Guidance (FFG)

The method used to compute FFG involves procedures opposite to those of a rainfall-runoff model. In other words, FFG is defined as the depth of rainfall over a given duration needed to initiate flooding at the outlet of a small stream basin. It is generally estimated for 1-, 3-, and 6-hour durations. In FFG, a specific amount of rain is required to produce a given amount of runoff based on estimates of current soil moisture conditions, which are derived from soil moisture models. Two quantitative products are needed to compute FFG: 1) threshold runoff and 2) rainfall-runoff curves.

The threshold runoff value represents an amount of excess rainfall over a given duration $tr$ required to induce flooding in small streams. Assuming that catchments respond linearly to excess rainfall, threshold runoff ($R$) can be estimated by

equating the peak catchment runoff determined from the catchment unit hydrograph over a given duration to the streamflow at the basin outlet associated with flooding which is expressed mathematically as follows:

$$Q_p = q_{pR} \times R \times A \quad \text{or} \quad R = \frac{Q_p}{A \times q_{pR}} \tag{1}$$

where $Q_p$ is the flood flow (cms or cfs), $q_{pR}$ is the unit hydrograph peak (cfs/mi$^2$/in) for a specific duration $tr$, A is the catchment area (km$^2$ or mi$^2$) and $R$ is the threshold runoff (cm or inches).

The flood flow $Q_p$ can be defined either physically as bankfull discharge $Q_{bf}$ or statistically as the two-year return period flow, $Q_2$. In this study, the threshold runoff criterion for small streams is a 0.5 m water level increase, as measured from the channel bottom, which is the level that mountain climbers and campers successfully escape from during natural flood damage. The discharge ($Q_{0.5wi}$) that causes a 0.5 m water level increase is defined. It was computed from channel geometry and roughness characteristics using Manning's formula for steady, uniform flow (Chow et al., 1988):

$$Q_{0.5wi} = B_{0.5wi} D_{0.5wi}^{5/3} S_c^{0.5}/n \tag{2}$$

where $B_{0.5wi}$ is the channel width at 0.5 m water level (m), $D_{0.5wi}$ is the hydraulic depth at 0.5 m water level (m), $S_c$ is the local channel slope (dimensionless), and $n$ is Manning's roughness. To obtain the peak catchment runoff, the unit hydrograph can be derived using various methods, such as Snyder's synthetic unit hydrograph approach (Chow et al., 1988) or the geomorphologic instantaneous unit hydrograph (GIUH) method (Rodríguez-Iturbe et al., 1979). In this study, we used the GIUH method to obtain peak catchment runoff.

To derive the rainfall-runoff curve which represents soil conditions during a flash flood event, it is necessary to estimate soil moisture. Soil moisture data are obtained via direct measurements with tensiometers or indirect methods such as rainfall-runoff models. In this study, the Sejong University Rainfall Runoff (SURR) model was used to estimate soil moisture. This model was developed based on the storage function model (SFM) (Kimura, 1961) and improved hydrological components such as potential evapotranspiration, surface flow, lateral flow, and groundwater flow based on the physical properties of these components (Bae and Lee, 2011). Moreover, this model uses estimates soil moisture continuously to determine time-dependent soil moisture conditions. The soil profile is separated into adsorbed water, tension water, and free water components. The soil water characteristics that distinguish these water components include the wilting point, field capacity, and saturated soil moisture conditions. The free water component in the soil profile contributes to lateral flow and percolation, while the tension water component contributes to actual evapotranspiration. Eq. (3) represents the soil water variations and hydrological component changes based on precipitation and potential evapotranspiration changes:

$$\frac{dSW(t)}{dt} = P(t) - AET(t) - Q_{sur}(t) - Q_{lat}(t) - Q_{gw}(t) \tag{3}$$

where $SW(t)$ is the soil water content (mm), $P(t)$ is the mean areal precipitation (mm) and $AET(t)$ is actual evapotranspiration (mm). $Q_{sur}(t)$, $Q_{lat}(t)$ and $Q_{gw}(t)$ denote the runoff components of surface flow (mm), lateral flow (mm), and groundwater flow (mm), respectively. Additional detailed mathematical descriptions of the components were

provided by Bae and Lee (2011). Bae and Lee (2011) showed that the SURR simulations are well fitted to observations, and Nash and Sutcliffe model efficiencies in the calibration and verification periods which are in the ranges of 0.81 to 0.95 and 0.70 to 0.94, respectively. Additionally, the behavior of soil moisture depending on the rainfall and the annual loadings of simulated hydrologic components are rational. From these results, an SURR model can be used for simulation of soil

moisture.

### 3.3 Receiver Operating Characteristics (ROC)

The Receiver Operating Characteristics (ROC) approach, or the ROC curve method, was originally proposed to analyse the classification accuracy associated with differentiating signals from noise in radar detection. This type of analysis is now widely used in several domains to assess the performance of statistical models that classify values into one of two categories.

A ROC curve plots the hit rate (*HR*) against the false alarm rate (*FAR*), which is computed using Eq. (4) and (5) and a contingency table or confusion matrix, as presented in Table 1. *H* and *M* represent hits and misses for predictions of when a flash flood will occur (OR > FFG). *F* and *N* represent false and negative hits for when a flash flood does not occur (OR < FFG).

$$Hit\ rate\ (HR) = \frac{H}{H+M} \tag{4}$$

$$False\ alarm\ rate\ (FAR) = \frac{F}{F+N} \tag{5}$$

Several contingency tables can be obtained based on varying decision thresholds associated with dichotomous events. The resulting point pairs (*FAR*, *HR*) from the contingency tables are plotted and connected by line segments. Additionally, they are connected to the point (0, 0), which corresponds to never forecasting the event, and to the point (1, 1), which corresponds to always forecasting the event. The perfect forecast yields values of *FAR*=0 and *HR*=1, i.e., the ROC curve consists of two

line segments that coincide with the left boundary and upper boundary of the ROC diagram. The upper left point of the graph represents perfect prediction. At the other extreme of performance forecasting, random forecasts based on sampled climatological probabilities can exhibit *FAR* = *HR*, and the ROC curve consists of a 45-degree diagonal line connecting the points (0, 0) and (1, 1). ROC curves that plot near the upper-left corner of the ROC diagram reflect better discrimination performance. Additionally, the area under a ROC curve can be used to summarize a ROC diagram, with the value of 1

representing a perfect forecast and 0.5 a random forecast. However, a ROC curve cannot be clearly indicated for objects that are more accurate than other objects. Wilk (2006) suggested an ROC Score which is the area of ROC curves. An ROC score can be calculated by using *HR* and *FAR*, as shown in Eq. (6) .

$$ROC\ Score = \left\{ \sum_{i=1}^{n} \frac{1}{2}(HR_i + 0.0)(FAR_i - 0.0) + \frac{1}{2}(HR_{i+1} + HR_i)(FAR_{i+1} - FAR_i) \atop + \cdots, + \frac{1}{2}(1 + HR_{i+n})(1 - FAR_{i+n}) \right\} \tag{6}$$

## 4 Results and Analysis

### 4.1 Regional regression relationships based on channel geometry

Threshold runoff values are based on the flood flow $Q_P$, unit hydrograph peak $q_{pR}$ and catchment area $A$. The discharge ($Q_{0.5wi}$) that causes 0.5 water level increase is used as a flood flow in this study. The calculations of $Q_{0.5wi}$ and $q_{pr}$ require

the channel cross-section parameters. Direct measurements of channel cross-sections, which are performed through local surveys, are not possible on a continuous spatial scale. Therefore, regional regression relationships are established between channel cross-section properties and the geometric characteristics of the upstream catchment to obtain cross-sectional information for un-surveyed streams. These regression relationships are established using stream survey data. The dataset includes channel width ($B$), hydraulic depth ($H$), and local channel slope ($S_c$) from on-site measurements. These data were

collected at 46 locations. Initially, the relationships between these parameters and the catchment area ($A$) were investigated using a power regression equation as follows:

$$X = \alpha A^{\beta} \tag{7}$$

where $X$ represents $B$, $H$ or $S_c$ and parameters α and β are determined by the regression of $X$ on $A$. Then, additional parameters such as stream length ($L$) and average basin slope ($S$) were investigated and included in the regression equation.

The regression relationship can then be expressed as follows:

$$X = \alpha A^{\beta} L^{\gamma} S^{\delta} \tag{8}$$

where $\alpha$, $\beta$, $\gamma$ and $\delta$ are regression coefficients. A correlation analysis was performed to analyse the relationship between the parameters ($B$, $H$, and $S_c$) and basin characteristics ($A$, $L$ and $S$). As shown in Table 3, the channel width $B$ was positively correlated with the catchment area $A$ but exhibited a significant negative correlation with the average basin slope $S$.

Conversely, the hydraulic depth $H$ was negatively correlated with $A$ but positively correlated with $L$ and $S$. The local channel slope $S_c$ was negatively correlated with $A$ and $L$. The derived regression equations are also shown in Table 3, and the determination coefficients of the regression equation were 0.76, 0.37 and 0.53 (Cho et al., 2011). The determination coefficient of hydraulic depth ($H$) is lower than the other variables. If additional data regarding river cross section are available, the regression equation will be improved.

### 4.2 Threshold runoff and FFG

The threshold runoff values were computed for effective rainfall durations of 1-hour in the 200 selected sub-basins by using Manning equation and GIUH method as mentioned in section 2.2. Figure 5(a) and (b) shows the estimated threshold runoff and its relative frequency in different ranges, respectively. Overall, the threshold runoff ranged from 18.7~42.8 mm/h with a mean of 31.8 mm/h. In addition, a large number of basins had threshold runoff values of 25~30 mm/h and 35~40 mm/h.

Figure 6 presents the soil moisture contents and deficits simulated using a continuous rainfall-runoff model, SURR, during the flooding season, i.e., July, August and September, from 2002 to 2009 for four selected sub-basins. In each figure, the

upper blue line represents the change in the soil moisture content based on the precipitation amount, while the grey dots represent the soil moisture deficit below saturation. The total soil moisture varied by sub-basin based on the soil conditions and basin characteristics. The soil moisture values were approximately 100-150 mm, 110 mm, 150 mm, 120 mm, and 105 mm in the Myungji, Soohang, Sanasa and Danjigol valleys, respectively. The soil moisture deficit generally ranged from 0~50 mm but was approximately 0~5 mm during 42% of the entire flood period. These values represent near-saturated soil conditions.

The mean area precipitation (MAP), estimated threshold runoff and FFG values for actual flash flood events that occurred in 2005, 2006, 2007 and 2009 in the Myungji, Soohang, Sanasa and Danjigol valleys, respectively, are presented in Figure 7. As shown in each figure, the values and trends of FFG, which is the sum of threshold runoff and the soil moisture deficit, differ by location. The values at Soohang valley and Sanasa valley are constant and indicate that the soil is already saturated due to antecedent precipitation, while the values at Myungji valley and Danjigol valley vary as precipitation inputs affect the soil moisture deficit. The time of flash flood occurrence was estimated based on when the hourly MAP exceeded the 1-hr FFG. Therefore, the time of flash flood occurrence was 0200 UTC on 3 August 2005 in the Myungji valley (32 mm MAP), 1300 UTC on 15 July 2006 in Soohang valley (66 mm MAP), 1600 UTC on 9 August 2007 in Sanasa valley (42 mm MAP) and 0600 UTC on 12 July 2009 in Danjigol valley (27 mm MAP). As shown in Table 2 and Figure 7, the timing of the flash flood occurrence computed from the FFG model exhibited satisfactory agreement with those from the observed flash flood record.

**4.3 Quantitative Threshold of Flash Flood Guidance**

Figure 8 shows the ROC scores of the four selected sub-basins estimated against virtual rainfall values of 1-100 mm/h with an interval of 1 mm/h. The virtual rainfall value associated with the peak ROC score was selected as the optimum rainfall. As expected, the minimum ROC score was 0.50, which represents random forecasting, while the maximum score and corresponding virtual rainfall were 0.90 and 32 mm/h in basin number 165, 0.91 and 30 mm/h in basin number 200, 0.87 and 22 mm/h in basin number 293, and 0.90 and 33 mm/h in basin number 442.

Similarly, the maximum ROC scores and corresponding optimum rainfall values were obtained in all other sub-basins. Figure 9 shows the ROC scores of all 200 sub-basins based on optimum rainfall values. The results show that the optimum rainfall values for flash flood warning criteria fall between 19 and 44 mm/h, with ROC scores of 0.85~0.98. An analysis of the selected optimum values and corresponding sub-basin areas revealed that the flash flood warning threshold could be best represented as a function of sub-basin area, as shown in Figure 10. Eq. (9) is a regression equation of a P-A curve that represents whether a flash flood will occur based on a given rainfall intensity and basin area:

$$P = 85.02 - 14.39 ln(A) \qquad (9)$$

where $A$ is the sub-basin area (km²) and $P$ is the hourly precipitation intensity (mm/h) that represent the quantitative flash flood criteria (QFFC). Thus, a flash flood will occur in a sub-basin with area $A$ if the rainfall intensity exceeds the P-A curve;

however, a flash flood will not occur if the rainfall intensity is below the curve. Note that the 1-hr precipitation intensity required to cause a flash flood decreases as a function of $A$.

In general, the P-A curve shows that a rainfall rate higher than 42 mm/h may trigger a flash flood in any sub-basin in the study area with an area greater than 22 km$^2$. We can further suggest the information of the flash flood threshold based on fieldwork in different sub-basins to refine the flash flood criteria. Flash flood warning thresholds were established for rainfall rates of 42 mm/h, 32 mm/h and 20 mm/h in sub-basins with areas greater than 20 km$^2$, between 40 and 100 km$^2$, and greater than 100 km$^2$, respectively.

## 4.4 Validation

For the validation of the performance of the P-A curve, the quantitative flash flood criteria for actual flash flood events were applied. This experiment assumed the gauged mean areal precipitation as a prediction. The experiments were assessed whether the prediction exceeded the quantitative flash flood criterion when an actual flash flood event occurred in the basins. If the prediction exceeded the quantitative flash flood criterion, a flash flood warning would be issued. According to the results, the flash flood occurrence was captured for 9 out of 12 events when the criteria were evaluated (Table 4). Figure 11 shows a detailed interpretation of the proposed criteria obtained from the P-A curve for the four selected actual flash flood events in the Myungji, Soohang, Sanasa and Danjigol valleys in 2005, 2006, 2007 and 2009. The 1-hr MAP and 1-hr criteria in the selected sub-basins with different areas were provided in the figures. The estimated values of 1-hr criteria were 31.9 mm, 37.2 mm, 37.7 mm and 31.7 mm for sub-basins area of 40.1 km$^2$, 27.8 km$^2$, 26.8 km$^2$, and 40.6 km$^2$, respectively. The 1-hr MAP exceeded the 1-hr criteria during the first three actual events, but the 1-hr MAP at Danjigol valley in 2009 event did not exceed the 1-hr criteria due to differences in the rainfall pattern and characteristics, as the precipitation distribution at Danjigol valley was continuous with double peaks, while those of other events were short periods with single peaks. Therefore, the flash flood occurrence at Danjigol valley was the effect of 3-hr cumulative rainfall rather than 1-hr rainfall. Thus, the flash flood occurred because the 3-hr maximum MAP (70.4 mm) was greater than the 3-hr FFG (67.5 mm). These results suggest that the proposed criteria derived from the P-A curve captured the flash flood occurrence effectively in each sub-basin.

## 5 Discussion

### 5.1 Uncertainty of flash flood forecasting method

There are many flash flood forecasting methods. The methods can be divided into three categories: flow comparison methods, rainfall comparison methods, and flash flood susceptibility assessment. The proposed P-A curve is rainfall threshold that included with the rainfall comparison methods like FFG. The rainfall comparison method is a popular tool for warning about flash floods, and this method is commonly used for flash flood forecasting. However, the previous rainfall threshold method has some limitations, recent studies tried to improve warning accuracy by using distributed physical

hydrological modeling (Kobold and Brilly, 2006; Reed et al., 2007; Norbiato et al., 2009). Hapuarachchi and Wang (2008) suggested that physically based distributed hydrological models are more appropriate than data-driven models and conceptual hydrological models for flash flood forecasting. However, the most important thing of flash flood forecasting is a providing the warning information to decision makers or citizens with relatively simple, clear, and immediate. It means that not only the sophistication but also promptness with reasonable accuracy also is necessary for flash flood forecasting. In this respect, this study proposed quantitative criteria using P-A curve for flash flood warning based on FFG. The key advantage of this method is that it doesn't need any further calculation compared to the other rainfall comparison method. In other word, the proposed criteria and methodology will serve as an important tool for issuing flash flood warnings based on only rainfall information.

However, this study has some assumptions and limitations. The P-A curve is based on the FFG, not real observed flash flood events because there is lack of observed flash flood events. In addition, the proposed P-A curve has some uncertainties from lots of sources such as soil moisture estimation (SURR), Threshold runoff estimation method, finding the optimal P-A curve by using ROC method, collection of actual flash flood events etc. But, these problems are not confined to this study because the phenomena triggering flash flood are very complex. Any flash flood forecast method has also large uncertainties due to input data errors, and modelling errors. Thus, it is necessary for understanding of the uncertainty from all these sources for decision making in flood warning because good uncertainty estimates of flash flood forecasts can add credibility to the forecast system.

## 5.2 Utilization of a P-A curve for flash flood forecasting

Some flood forecasting systems have been developed and operated in some countries (Mogil et al., 1978; Sweeney, 1982; Mason, 1982; Alfieri et al., 2012). Northern America has a flash flood forecasting system using gridded flash flood guidance (GFFG). This system uses multi-sensor precipitation estimates and forecasts based on NEXRAD (Next Generation Weather Radar), rain gauges and NWP (numerical weather prediction) model outputs. The European Flood Forecasting System (EFFS) used the LISFLOOD-FF for generating river flow and LISFLOOD-FP to model the overbank flows and inundation areas, and they use gauged rainfall, radar rainfall, and NWP model outputs (Roo et al., 2003). ALERT in Australia uses a hydrological model with real-time rainfall and water level data. They also assess the severity of flooding using simple manual guides (look-up tables). Thus, the ideal flash flood system needs to combine two approaches. It must present the criteria used to judge flash floods in an intuitive way for very short-term flash floods (less than 1 hour). It must also make predictions with sophisticated modeling using a physical distributed model for flash floods with greater than a 3-hour duration. Therefore, the FFGC (flash flood guidance criteria) are used for short-term flash floods.

This study focused on using a P-A curve, and it assessed the outcome when using only gauged rainfall data. However, the quality of flash flood forecasting depends on the quality of the rainfall data. Additionally, reliable rainfall forecasts with adequate lead-time and accuracy are essential for flash flood forecasting. In general, the gauged rainfall, radar data (Sinclair and Pegram, 2005; Mazzetti and Todini, 2009), and satellite data (Sooroshian et al., 2000; Kubota et al., 2007) have been

used for quantitative precipitation estimates (QPEs), and some studies have used multiple precipitation sources (Sokol, 2006; Chiang et al., 2007). Therefore, this method is necessary for assessing the applicability of using rainfall data obtained from various sources.

## 6 Conclusion

In this study, quantitative criteria for flash flood warning were developed and assessed for sub-basins of the Han River in South Korea. Flash flood guidance based on threshold runoff was estimated for 200 sub-basins. The optimum rainfall values were obtained for each sub-basin by comparing FFG, virtual rainfall and observed rainfall values using a ROC analysis. The optimal rainfall values for the flash flood warning threshold were between 19 and 44 mm/h, with a ROC score of 0.85–0.98. The flash flood warning threshold can be best represented as a function of sub-basin area. A generalized precipitation–area curve of $P = 85.02 - 14.39\,ln(A)$ was proposed to the Han River basin in South Korea. The results showed that it could be effectively estimated as a function of the corresponding sub-basin area. These results mean that the threshold for 1-hr flash flood prediction can be classified according to sub-basin area.

The key advantage of this method is possible to issue flash flood warnings without the need to run entire hydro-meteorological model chains in the region where the flash floods with less than1-hr duration are frequently occurred. However, flash flood with more than a 3-hr duration maybe sensitive to the soil moisture condition, and have response time. Therefore, the development of the coupled flash flood forecasting system which is divided with short (less than 1 hr) and long-duration (greater than 3 hrs) is necessary for managing flash flood efficiently.

## Acknowledgements

This work was supported by a subproject of the Development of HPC-based management systems against national-scale disasters project and supported by the Korea Institute of Science and Technology Information (KISTI), and the National Research Foundation of Korea (NRF) grant funded by the Korean government (MSIP) (no. 2011-0030040).

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

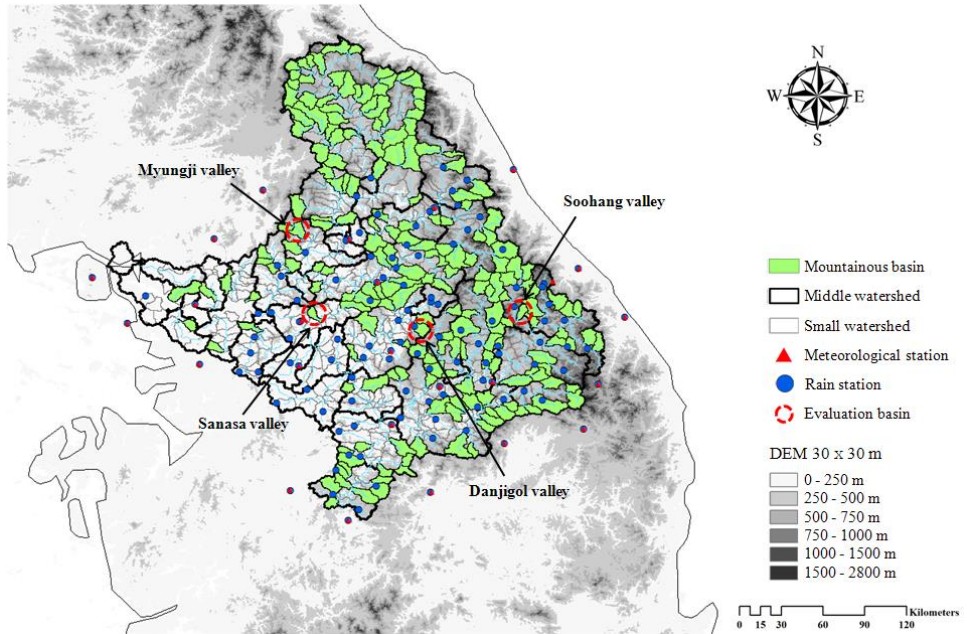

**Figure 1: Study area.**

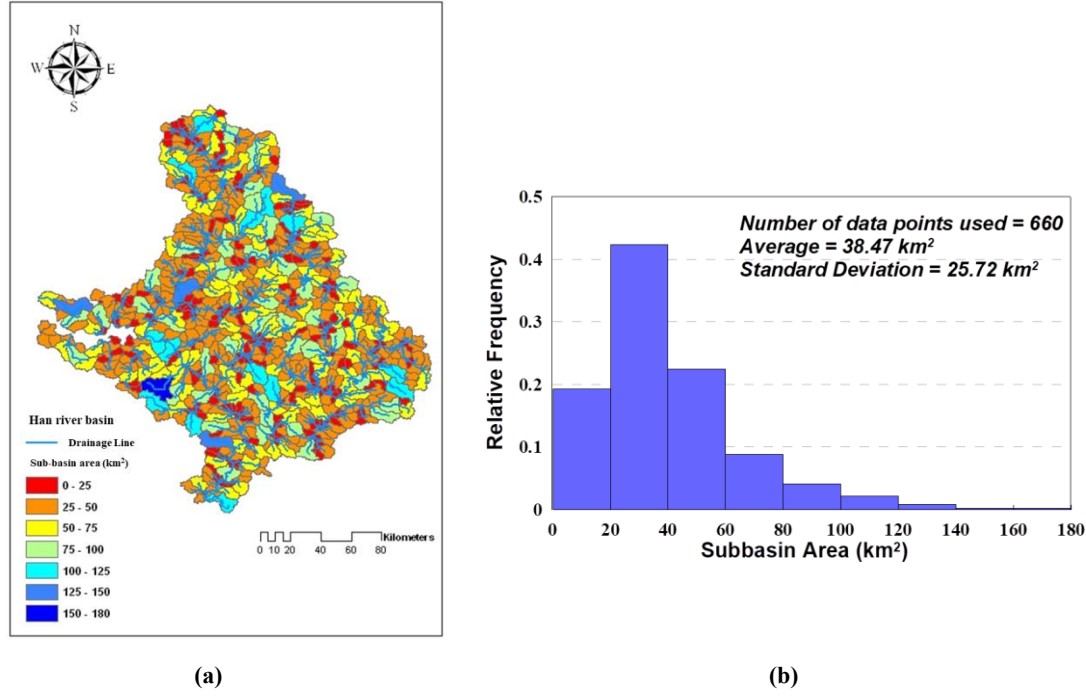

(a)                                             (b)

**Figure 2: (a) 660 sub-basins in the Han River basin and (b) their relative frequency of their areas.**

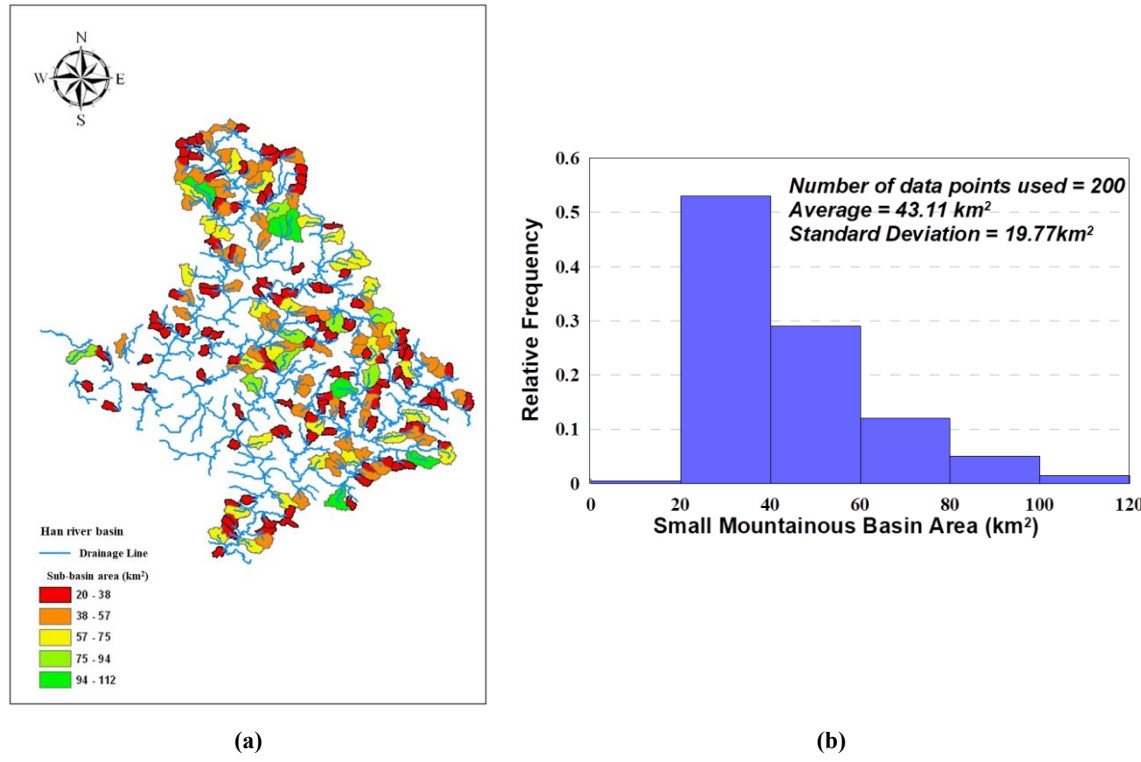

(a)                                                    (b)

**Figure 3: (a) Selected 200 sub-basins in the Han River basin and (b) the relative frequency of their areas.**

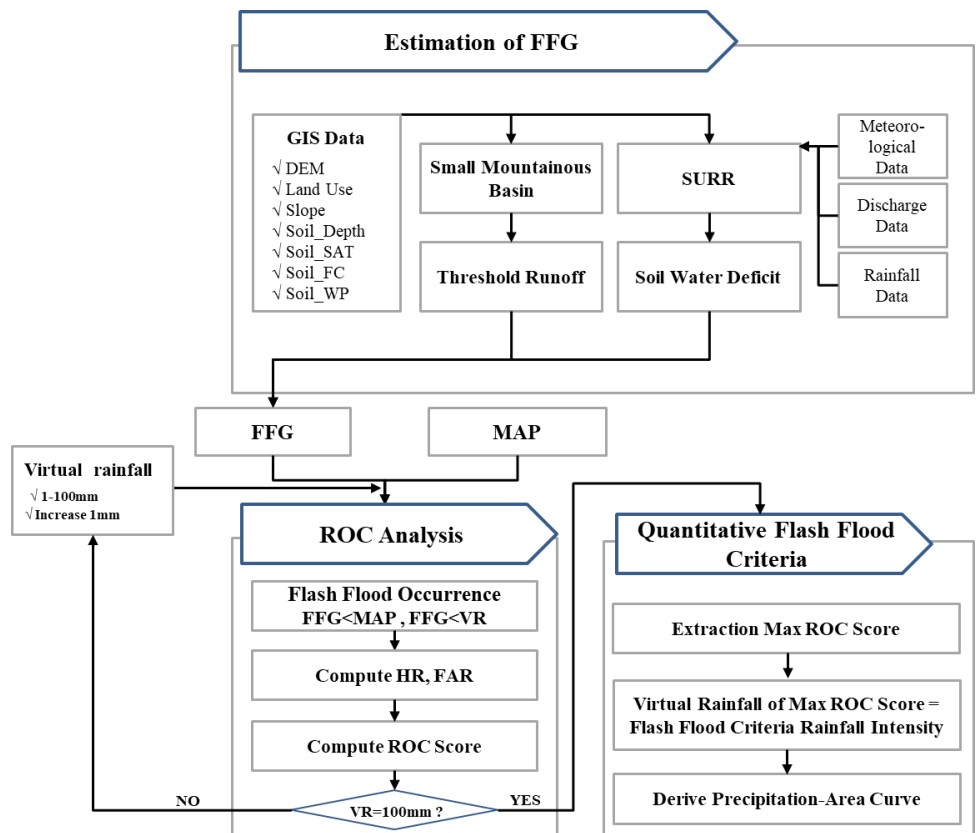

**Figure 4: Overall methodology used to estimate the quantitative precipitation criteria.**

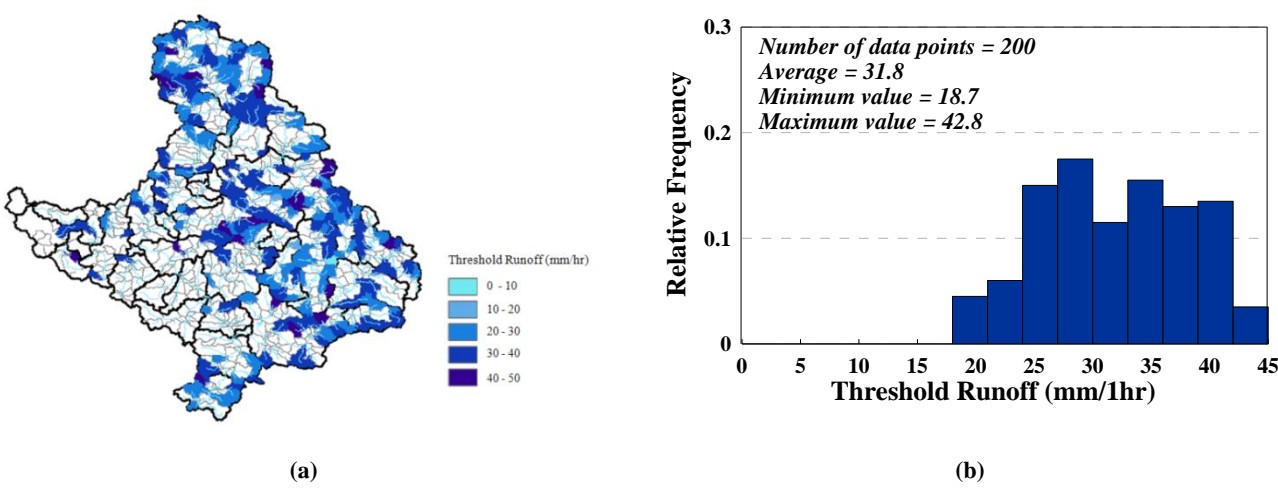

**(a)**                                                       **(b)**

Figure 5: (a) Threshold runoff and (b) the relative frequency of threshold runoff.

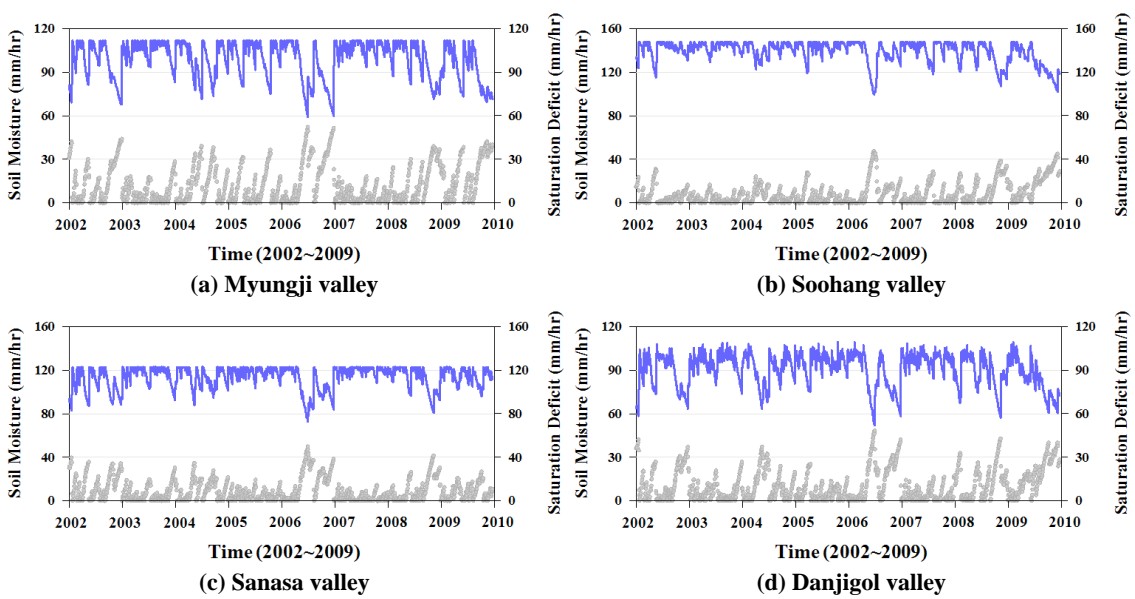

**Figure 6: Soil moisture and soil moisture deficit in selected sub-basins. The blue and grey lines represent soil moisture, and saturation deficit, respectively.**

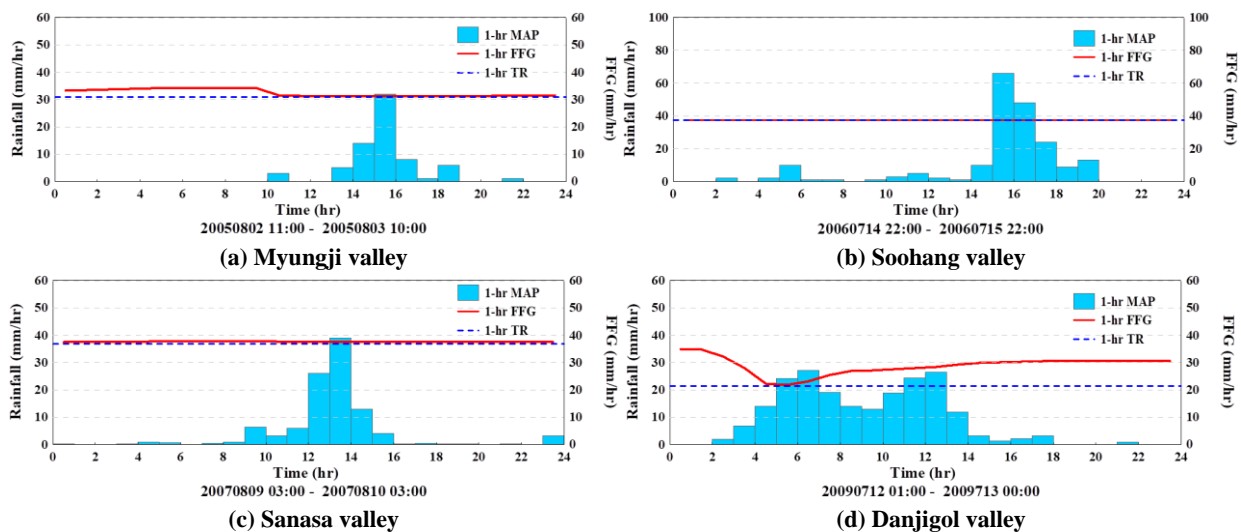

**(a) Myungji valley**

**(b) Soohang valley**

**(c) Sanasa valley**

**(d) Danjigol valley**

**Figure 7: Threshold runoff (TR), Mean areal precipitation (MAP) and estimated FFG (Flash Flood Guidance) for selected flash flood events.**

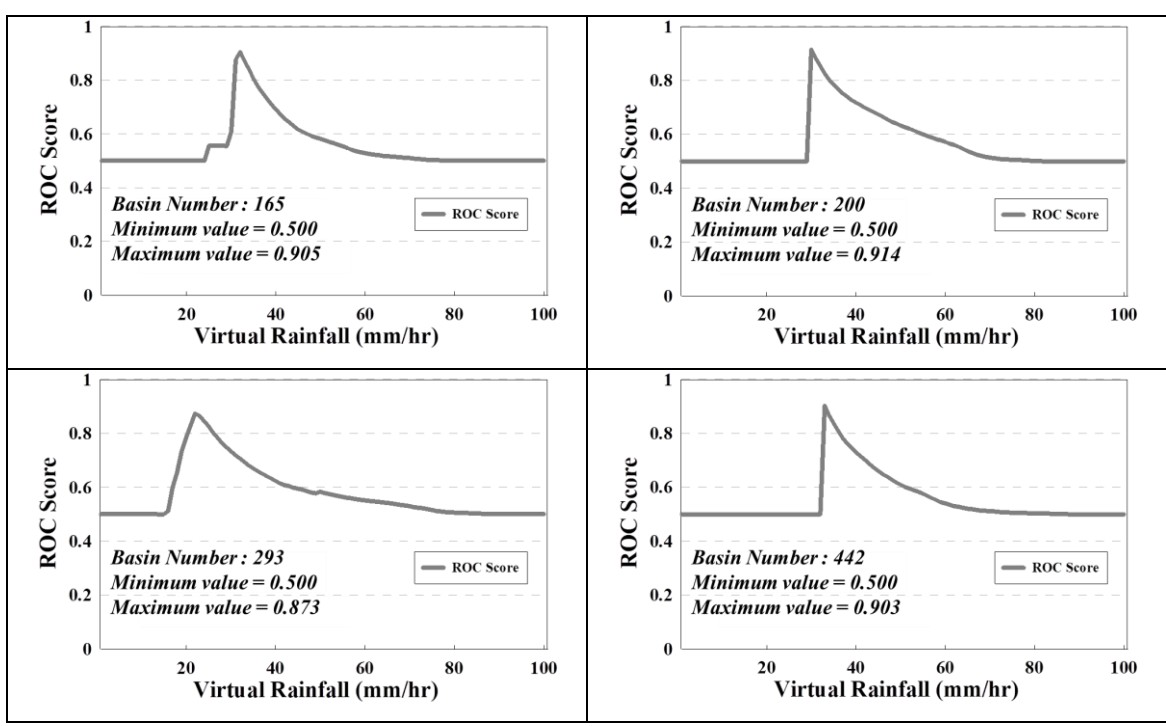

**Figure 8: ROC score estimated for selected sub-basins using virtual rainfalls of 1–100 mm/h.**

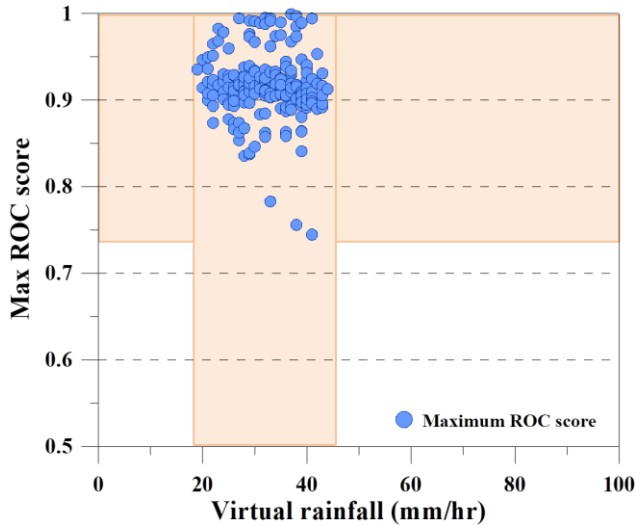

**Figure 9: Relationship between maximum ROC and uniform virtual rainfall for all the sub-basins. Shaded areas represent the range (maximum to minimum) of virtual rainfall and ROC score.**

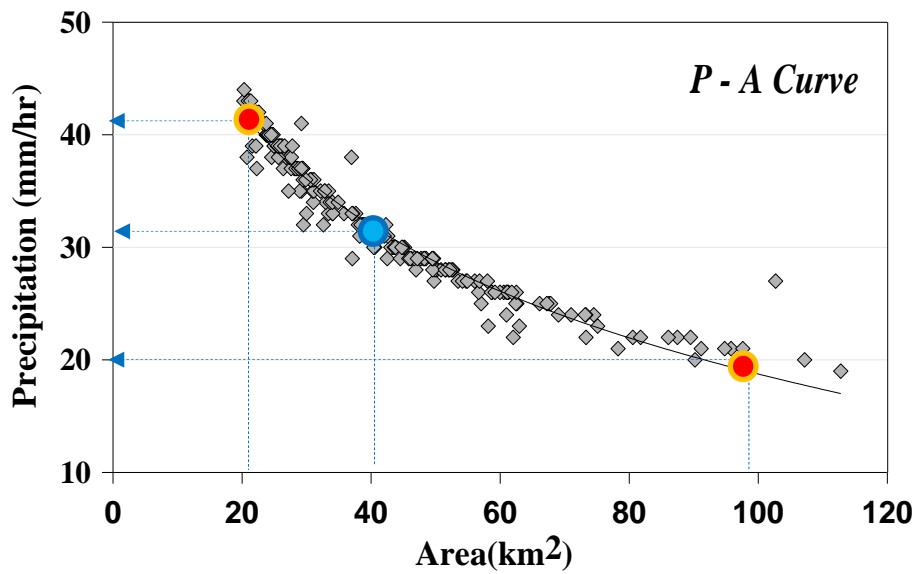

**Figure 10: Derived QPC curve for quantitative flash flood conditions  Circles represent the categories of criteria according to basin area**

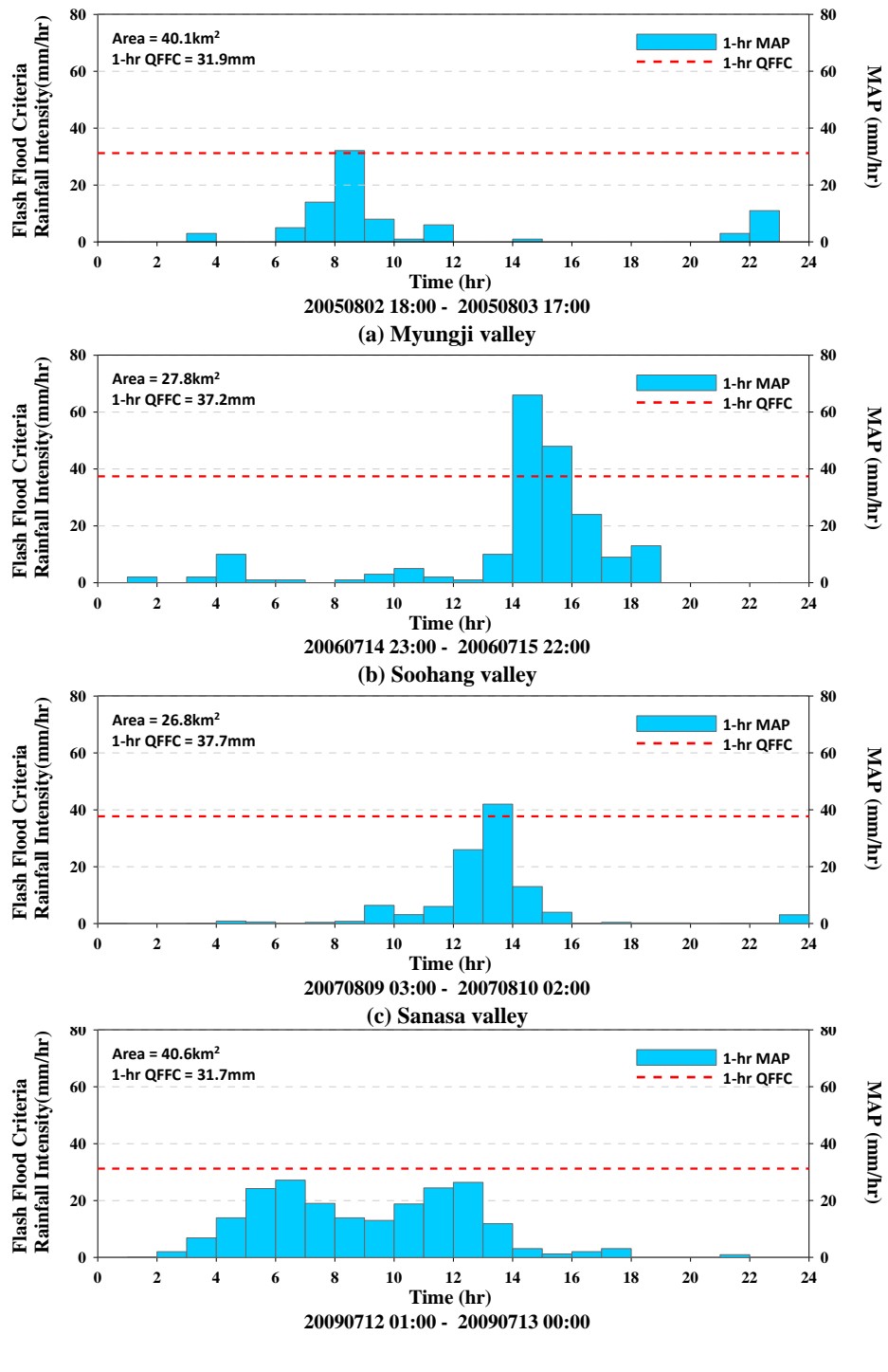

**Figure 11: Validation of quantitative flash flood criteria.**

**Table 1: Flash flood records collected in the Han River basin.**

| S. No. | Time | Location | Area (km$^2$) | Longitude | Latitude | Maximum rainfall (mm/hr) |
|---|---|---|---|---|---|---|
| 1 | 2005.08.03 02:00 | Mt. Myungji valley, Gapyeong-gun, Gyeonggi-Do | 40.06 | 37.9447 | 127.4949 | 32.1 |
| 2 | 2006.07.15 13:00 | Soohang valley, Pyeongchang-gun, Gangwon-Do | 27.79 | 37.5619 | 128.6087 | 66.0 |
| 3 | 2007.08.09 16:00 | Sanasa valley, Yangpyeong-gun, Gyeonggi-Do | 26.83 | 37.5353 | 127.5292 | 42.0 |
| 4 | 2009.07.12 06:00 | Danjigol valley, Hoengseong-gun, Gangwon-Do | 40.58 | 37.3974 | 128.1232 | 27.2 |
| 5 | 2010.09.11 19:00 | Yongchoo valley, Gapyeong-gun, Gyeonggi-Do | 44.92 | 37.8561 | 127.4832 | 37.0 |
| 6 | 2011.07.27 05:00 | Uidong valley, Gangbuk-Gu, Seoul-Si | 36.96 | 37.6711 | 127.0060 | 57.5 |
| 7 | 2011.07.26 17:00 | Madangbawii valley, Namyangju-Si, Gyeonggi-Do | 46.95 | 37.7039 | 127.1008 | 49.2 |
| 8 | 2011.07.27 08:00 | Mt. Namhan valley, Gwangju-Si, Gyeonggi-Do | 33.99 | 37.4786 | 127.1887 | 52.1 |
| 9 | 2011.08.03 12:00 | Noksoo valley, Gapyeong-gun, Gyeonggi-Do | 37.63 | 37.7764 | 127.3954 | 38.1 |
| 10 | 2011.08.09 13:00 | Sadam valley, Goesan-gun, Chungbuk | 60.98 | 36.6253 | 127.8312 | 22.1 |
| 11 | 2011.08.14 13:00 | Gogiri valley, Yongin-Si, Gyeonggi-Do | 40.45 | 37.3599 | 127.0560 | 23.2 |
| 12 | 2012.07.15 08:00 | Byeongjibangri valley, Gapyeong-gun, Gyeonggi-Do | 90.18 | 37.6080 | 128.0762 | 21.5 |

**Table 2: ROC analysis for quantitative precipitation criteria.**

| | | Observed event | |
|---|---|---|---|
| | | Positive (OR>FFG) | Negative (OR<FFG) |
| Virtual event | Positive (VR>FFG) | Hit (H) | False (F) |
| | Negative (VR<FFG) | Missing (M) | Negative hit (N) |

**Table 3: Regression analysis for parameter estimation using basin area, stream length and slope in the Han River basin.**

| Parameter | Best-fit regression | Coefficient of determination, $R^2$ | No. of cases |
|-----------|--------------------|------------------------------------|--------------|
| B | $= 15.776 A^{0.369} S^{-0.0080}$ | 0.76 | 46 |
| H | $= 2.39 A^{-0.920} L^{1.174} S^{0.748}$ | 0.37 | 46 |
| Sc | $= 2.443 A^{-0.278} L^{-0.769}$ | 0.53 | 46 |

Units: B [ft], H [ft], S [ft/mi], Sc [ft/mi], A [mi$^2$], and L [mi]

**Table 4. Validation of FFGC using observed flash flood cases**

| S.No. | Time | Area (km$^2$) | MAP (mm/hr) | FFGC (mm/hr) | FF occurrence using FFGC |
|---|---|---|---|---|---|
| 1 | 2005.08.03 02:00 | 40.1 | 32.1 | 31.9 | ◯ |
| 2 | 2006.07.14 13:00 | 27.8 | 66.0 | 37.2 | ◯ |
| 3 | 2007.08.09 16:00 | 26.8 | 42.0 | 37.7 | ◯ |
| 4 | 2009.07.12 06:00 | 40.6 | 27.2 | 31.7 | x |
| 5 | 2010.09.11 19:00 | 44.9 | 37.0 | 30.3 | ◯ |
| 6 | 2011.07.27 05:00 | 37.0 | 57.5 | 33.1 | ◯ |
| 7 | 2011.07.26 17:00 | 47.0 | 49.2 | 29.6 | ◯ |
| 8 | 2011.07.27 08:00 | 34.0 | 52.1 | 34.3 | ◯ |
| 9 | 2011.08.03 12:00 | 37.6 | 38.1 | 32.8 | ◯ |
| 10 | 2011.08.09 13:00 | 61.0 | 22.1 | 25.9 | x |
| 11 | 2011.08.14 13:00 | 40.5 | 23.2 | 31.8 | x |
| 12 | 2012.07.15 08:00 | 90.2 | 21.5 | 20.2 | ◯ |