# Peer review of "Development of a Precipitation-Area Curve for Warning Criteria of Short-Duration Flash Flood"

_Natural Hazards and Earth System Sciences, 2017_

## Referee Comment (RC1) · Anonymous Referee #1 · 24 Jul 2017

Review of

'Development of a Precipitation-Area Curve for Warning Criteria of Short-Duration Flash Flood' submitted to NHESS by Deg-Hyo Bae, Moon-Hwan Lee, Sung-Keun Moon

Dear Authors,

In this manuscript you propose a model for predicting (and warning of) flash floods in parts of South Korea. You argue that a soil-water model is an important component for better deriving conditions that lead to flash floods. You also include the concepts of flash flood guidance (FFG), threshold runoff, and simulations of virtual rainfall to arrive at a precipitation-basin area curve that helps predict flash floods. The backbone of your approach seems to be a classifier that allows you to 'predict' flash floods from

time series. One could interpret your precipitation-area curve as a decision boundary, although you do not explicitly investigate this concept.

Your topic is clearly of interest to NHESS and a broad international readership, but the way you present your research needs very thorough attention. I suggest restructuring your manuscript, better outlining your methods and assumptions, adding a dedicated discussion section, and carefully revisiting your concept of validation. You could also help readers appreciating the novelty and advances of your contributions by more clearly and critically assessing what you have achieved here.

__General Remarks__

–Your abstract could do with more detail on how you validated your predictions, and whether they are reliable enough to allow useful flash-flood predictions (or forecasts). What is the eventual output of your prediction and where can this be used in practice?

–The introduction provides some clues why forecasting flash floods is important, but misses opportunities to briefly explain those concepts (especially 'FFG') relevant to your research. Consider making a better case by illuminating more recent case studies of flash floods in South Korea. What is mostly needed for their prediction and why? In this regard, you close the introduction with a somewhat contradictory comment on the need for measuring (antecedent?) soil moisture. Please reconcile that statement and offer a clear overview of your objectives. Which research question is it that you wish to address? Which tools do you use and why?

–The methods section I found difficult to follow. You start of with QPC computation and briefly mention the concept of 'virtual rainfall'. Please elaborate more on that so that readers can reproduce the full stream of your methods. Provide (more) mathematical formulations where appropriate, and please do explain all parameters used (some are not referred to). Why use bankfull discharge? Is that the definition for the minimum discharge to cause flash flooding? Assuming steady, uniform flow may also be problematic for flash floods, and you might want to pick that up in the discussion. Equation
3 shows a soil-water content balance that you adapt from the SURR model; how well can you constrain each of the five terms? For example, will evapotranspiration as a function of time be relevant for forecasting flash floods? Clearly you want to specify the timescales that you base your forecasts on. I was unsure about the output of your model. Your use of a receiver-operating-characteristic curve indicates that you classify something, but what exactly, remains vague. Please explain in more detail how you labeled the classes of observed events and how you predicted new classes using SURR.

–I suggest changing the order of the methods and study area section. Providing first a general background on the region of interest and the data available before dealing with the method makes more logical sense to me.

–The results section starts off with more methods, uncomfortably emphasizing even more the logical disruption between the early sections of your manuscript. You offer some hydraulic geometry that you derive from a multiple regression model, in which the predictors are clearly correlated. This will need some more robust statistical treatment. Further down the section you mention that the predicted and observed timing of flash floods seem to be roughly similar. This is the first explicit mention of comparing predictions with observed data, and thus the motivation for using ROC curves, I presume. If so, please make sure that this core message comes across much earlier. Again, the time steps or measurement/simulation intervals here are critical. Please elaborate. I am a bit suspicious about Fig. 10. Does basin area somehow play a role in estimating rainfall intensity in any of your models? Finally, your validation (section 4.4) needs to be more convincing. You mention that you tested your method on four observed flash floods between 2005 and 2009. How many cases did you use for training your classifier? Can you show some ROC curves (or other performance metrics) for the testing cases?

–Your study could use a formal discussion section, in which you objectively discuss your methods in the light of their assumptions, limitations, and benefits (or advances)

compared to previous work. Consider reflecting on how accurately SURR produces the necessary input data; how your classification would change for different time intervals; how your classification deals in general with rare events (for which ROC curves might not be the best of performance metrics); and what you consider as possible future improvements to your model.

–Your conclusions mostly summarize your data. You report a high prediction potential, which is partly based on finding the optimal ROC scores in the first place, right? You state that 'The flash flood warning threshold can be best represented as a function of sub-basin area' (page 8/line 27). What does that mean and what is its practical relevance for warning? You may want to report statistical uncertainties for your generalized precipitation-area in this context.

–The reference list appears a bit short. I imagine that other groups must be working on prediction of flash floods elsewhere.

–Figures: #1 is OK, if you add some explanatory detail to the caption; please explain all abbreviations. #2 needs geographic coordinates and larger fonts. #3a and # 4a need units for 'sub-basin area'; are #3b and #4b really necessary? Histogram bins in #5b may be too wide: what is it that you wish to state here? #6 needs explanations of color codes. #7 needs larger fonts and explanation of abbreviations. #8: it is unclear what the minimum and maximum numbers refer to. #9: please explain orange shades. #10: please explain red and blue circles. Overall, you may want to use your captions for informing readers more about the contents and messages of your figures.

–Is Table 1 necessary?

–Please ask a native speaker to check your manuscript. I have noticed numerous formal and potentially ambiguous errors in the text, but these errors are too many to list in detail below. Therefore I only a give only a few examples in the line-specific suggestions below.

__Specific Suggestions (page/line)__

1/8: Delete 'with short duration'. The term 'flash flood' implicates that.

1/9: 'required to cause minor flooding' - Why minor flooding? Please provide a brief definition of what you mean by 'minor' here.

1/12: Please spell out 'ROC'.

1/15: 'highly' should read 'more'?

1/16: 'obtained for rainfall rates of 42, 32 and 20 mm/h' - It is unclear why or how you picked those rates. Please explain.

1/17: 'actual' means 'observed' or 'measured'? Please summarize briefly the results from your validation.

1/20: 'the short-duration flash flood frequently occurred' should read 'the flash floods occur frequently'.

1/24: 'managing flash flood control' - What do you mean by that specifically?

1/25: 'the climate change has increased' could read 'climate change may have likely increased'.

1/27: What sort of 'technology' do you mean? Or did you mean 'methodology' instead?

1/28: 'For deciding flash flood occurrence,' - Unclear.

2/1: 'flash flood vulnerability' - This refers to potential damage. Is that what you meant?

2/5: 'simulation to establish the observed frequency distribution' - Contradictory. Why simulate something to establish observations? Perhaps change the wording here?

2/6: 'comparing forecast flow with flooding flow' - How about 'comparing forecast with observed flows'?

2/7: Delete 'eminent'.

2/9: 'understood by the general public' - It may be useful to briefly explain the concept here.

2/13-15: So what did those studies find out?

2/20: 'the hourly maximum rainfall exceeded 50mm/hr and 60 mm/hr in 2006 and 2011' - Difficult to assess the relevance of these rates without any background information on rainfall characteristics in the region.

2/23: Delete 'exquisitely'. Please also check grammar in this sentence. I think I know what you mean here, but you would be really well advised to seek the help of a native speaker for rephrasing many similar statements in your manuscript.

3/9: What are 'ROC scores'?

3/15: 'method used to compute FFG is the opposite' - So what is the main output of FFG?

3/19: 'over a given duration tr required to' - Please use italics for all parameters that you introduce.

3/24: What is the unit of the 'unit hydrograph peak', if you use differing metric systems? Please attend to Equation 1: in my copy of the PDF it looks as if A is an exponent in the denominator.

4/4: 'which represents current soil conditions' - What do you mean by 'current'? During or before the flash flood?

4/9: 'this model uses estimates soil moisture' - Ambiguous. Does the model use estimates of soil moisture or does it estimate soil moisture itself? That is a big difference.

4/27: Please explain parameters in Equations 4 and 5.

5/5: 'line segments that coincide with the left boundary and upper boundary of the ROC diagram' - You could simply say that, for a perfect prediction, the ROC curve has

to pass through (0, 1) or the upper left point of the graph.

5/7: 'ROC curves associated with real forecasts generally fall between these two extremes and plot above and to the left of the 45-degree diagonal' - Not sure what you mean by a 'real' forecast.

5/15: 'were delineated' - How did you delineate those basins? Their size spans three orders of magnitude, so what was the underlying rationale?

5/19: 'omitted from further analysis' - So you did not consider all basins with reservoirs further?

5/21: 'filtering' - This means you had some preconception about basin area influencing flash-flood potential? It might be good to give more detail here.

5/28: 'soil moisture conditions were estimated' - Please be more specific about the spatial resolution, time intervals, and accuracies of those estimates.

5/30: 'flood information was obtained through different sources, including print and electronic media' - How homogeneous and reliable is that information?

6/1: 'multiple flash flood events' - Perhaps this is something you may wish to elaborate on a bit more?

6/15: 'were investigated and included in the regression equation' - Please describe this in more detail. You note that some of the predictors in your regression model are correlated, but you do not seem to do anything about this.

6/25: 'Threshold runoff values were computed' - How?

6/27: Can you measure runoff rate to one tenth of a mm/h?

6/30: 'flooding season, i.e., July, August and September' - You could explain more about this flooding season in the study area descriptions; international readers might welcome this information.

7/16: 'times of flash flood occurrence computed from the FFG model exhibited satis-factory agreement' - Is it the timing that you wish to classify correctly?

7/21: 'As expected, the minimum ROC score was 0.50' - You can sometimes get lower values than that.

8/12: 'estimated values of 1-hr QFFC' - Do you have measured values for a validation?

8/29: 'optimum threshold for flash flood warning in a sub-basin' - Slight repetition.

9/4: 'which is divided with short and long-duration' - And how do set the threshold between 'short' and 'long'?

---

## Referee Comment (RC2) · Anonymous Referee #2 · 4 Sep 2017

Important topic of high relevance. The paper is generally of a good structure and gives good insight in what has been done in the project. But not really a new approach, already been done in similar ways in other regions. Following open issues should be addressed in the publication: The cells of convective events are much smaller than the catchments described in the paper, so in the real world only part of the catchment will be in the focus of the precipitation event. There is little info about what type of precipitation measurement has been used, are this ground measurements or radar or some combination of it? What is the resolution of the measurement? As the convective events are difficult to measure, the uncertainty applied by this also should be discussed. In the publication a lot of abbreviations are being used, this makes it hard to read, especially as they are not commonly used abbreviations, so better replace them

by the full text. As the timely distribution of a rainstorm event also has high impact on the runoff, this should be tackled as well. Some more words should be spend on how the results can be used, in what extent are the usable for warning issues and how false warnings can be handled. Who is the planned end user of the thresholds?

---

## Author Comment (AC1) · 30 Oct 2017

Revision according to the reviewer's comments:

We thank you for your constructive review and comments. We have attached answers and the revised manuscript. The comments are numbered by reviewer comment; our responses were written after this symbol (-). We believe that the manuscript is in much better form now.

Reviewer:

In this manuscript you propose a model for predicting (and warning of) flash floods in parts of South Korea. You argue that a soil-water model is an important component for better deriving conditions that lead to flash floods. You also include the concepts

of flash flood guidance (FFG), threshold runoff, and simulations of virtual rainfall to ar-
rive at a precipitation-basin area curve that helps predict flash floods. The backbone
of your approach seems to be a classifier that allows you to 'predict' flash floods from
time series. One could interpret your precipitation-area curve as a decision boundary,
although you do not explicitly investigate this concept. Your topic is clearly of interest
to NHESS and a broad international readership, but the way you present your research
needs very thorough attention. I suggest restructuring your manuscript, better outlining
your methods and assumptions, adding a dedicated discussion section, and carefully
revisiting your concept of validation. You could also help readers appreciating the nov-
elty and advances of your contributions by more clearly and critically assessing what
you have achieved here.

-We also agree with your opinion that the submitted manuscript did not fully describe
the novelty and advances of study. And there are several ambiguous expressions in
the manuscript. Therefore we tried to show the obvious motivation, purpose, and final
output of this study in abstract, introduction sections. We added the discussion section
for suggesting the meaning, limitation, utilization, and future work of this study. And we
rephrased the methodology and results sections for removing ambiguous expression
and providing more information with readers.

*General Remarks

1. Your abstract could do with more detail on how you validated your predictions, and
whether they are reliable enough to allow useful flash-flood predictions (or forecasts).
What is the eventual output of your prediction and where can this be used in practice?

-We added detailed description of methods and key results of validation, indicating
the usefulness of the P-A curve in practice (page 1, line 17-20). "The proposed P-
A curve was validated based on observed flash flood events in different sub-basins.
Flash flood occurrences were captured for 9 out of 12 events. This result can be
used instead of FFG to identify brief flash flood (less than 1-hour), and it can provide

warning information to decision makers or citizens that is relatively simple, clear, and immediate."

2. The introduction provides some clues why forecasting flash floods is important, but misses opportunities to briefly explain those concepts (especially 'FFG') relevant to your research. Consider making a better case by illuminating more recent case studies of flash floods in South Korea. What is mostly needed for their prediction and why? In this regard, you close the introduction with a somewhat contradictory comment on the need for measuring (antecedent?) soil moisture. Please reconcile that statement and offer a clear overview of your objectives. Which research question is it that you wish to address? Which tools do you use and why?

-We added the some explanation of the relationship between FFG and the P-A curve in the methods section (page 2, line 33∼page 3, line 3). "Although FFG-based methods provide useful mechanisms for flash flood warning, the real-time estimates of soil moisture required in some regions are often challenging to acquire prior to rapid response against flash floods. In this study, we proposed quantitative criteria using a P-A curve for flash flood warning based on FFG due to the lack of observed flash flood events. Thus, a P-A curve was derived by using FFG, but we validated the criteria by using observed flash flood events"

-We added a literature review related to flash flood studies in South Korea, and we revised the introduction to suggest the research questions and purposes of this study (page 2, line 18 ∼30). "Bae and Kim (2007) provided the flash flood guidance using the Manning equation, GIUH (geomorphologic instantaneous unit hydrograph), and TOP-MODEL (Beven et al., 1994). Lee et al. (2016) generated a gridded flash flood index using the gridded hydrologic components of the TOPLATS land surface model and a statistical flash flood index model. Recent studies have focused on the accuracy and spatial distribution of FFG. However, South Korea has recently suffered many flash flood events in the mountainous regions. More than 64% of South Korea is mountainous and prone to flash floods with very short rainfall durations. Recent heavy rainfalls

in South Korea have triggered flash floods and landslides that caused severe damage to infrastructure and resulted in dozens of deaths. Notably, the heavy rainfall events have resulted in several flash floods since 2000, such as events in 2005, 2006, 2008 and 2012 at several locations in South Korea. In particular, the hourly maximum rainfall exceeded 50 mm/hr in 2006 and 2011, most of the flash flood events in South Korea were caused by short rainfall duration of less than one hour. It is difficult to capture these flash flood cases using the methods presented in previous studies. Therefore, prompt flash flood warnings are necessary for citizens and decision-makers."

-We added some reasons to use FFG (page 3, line 1-3). "In this study, we proposed quantitative criteria using a P-A curve for flash flood warning based on FFG due to the lack of observed flash flood events. Thus, a P-A curve was derived by using FFG, but we validated the criteria by using observed flash flood events."

-And we describe our reasons for using discharge that causes a 0.5m water level increase and SURR in section 3 (page 5, line 4~6; page 5; line 29~ page 6, line 1). "In this study, the threshold runoff criterion for small streams is a 0.5 m water level increase, as measured from the channel bottom, which is the level that mountain climbers and campers successfully escape from during natural flood damage. The discharge ($Q\_0.5wi$) that causes a 0.5 m water level increase is defined." "Bae and Lee (2011) showed that the SURR simulations are well fitted to observations, and Nash and Sutcliffe model efficiencies in the calibration and verification periods which are in the ranges of 0.81 to 0.95 and 0.70 to 0.94, respectively. Additionally, the behavior of soil moisture depending on the rainfall and the annual loadings of simulated hydrologic components are rational. From these results, an SURR model can be used for simulation of soil moisture."

3. The methods section I found difficult to follow. You start of with QPC computation and briefly mention the concept of 'virtual rainfall'. Please elaborate more on that so that readers can reproduce the full stream of your methods. Provide (more) mathematical formulations where appropriate, and please do explain all parameters used (some

are not referred to). Why use bankfull discharge? Is that the definition for the minimum discharge to cause flash flooding? Assuming steady, uniform flow may also be problematic for flash floods, and you might want to pick that up in the discussion. Equation 3 shows a soil-water content balance that you adapt from the SURR model; how well can you constrain each of the five terms? For example, will evapotranspiration as a function of time be relevant for forecasting flash floods? Clearly you want to specify the timescales that you base your forecasts on. I was unsure about the output of your model. Your use of a receiver-operating-characteristic curve indicates that you classify something, but what exactly, remains vague. Please explain in more detail how you labeled the classes of observed events and how you predicted new classes using SURR.

-We revised Figure 1 to provide a more detailed description and added some explanation (page 4, line 5~12). This revision makes it clearer how we apply the ROC analysis and how we derive the P-A curve.

"This study presents a method for deriving a P-A curve that represents the rainfall thresholds occurring during flash floods. The method is based on FFG analysis to avoid the need to estimate soil moisture conditions. Figure 4 presents the overall procedure used to evaluate the quantitative precipitation criteria (QPC) for flash flood warning. First, the mean areal precipitation and FFG were calculated by using topographic, meteorological data for the sub-basins in the study area. To obtain FFG at current time (t), which is a summation of threshold runoff (TR) and soil moisture deficit, threshold runoff at each sub-basin is estimated. The soil moisture conditions from actual rainfalls are simulated by using SURR model, and we can decide whether a flash flood occurred at certain basin by comparing this FFG value and that from 1-hr prior to the actual rainfall. In this experiment, it is assumed that if the observed MAP is larger than the FFG, a flash flood occurs."

-We added explanations of all parameters in the equation. -We used discharge at the level of a 0.5 m water level increase from the channel bottom which is the level from

which mountain climbers and campers can successfully escape during natural flood damage. -We chose the SURR model because this model can simulate continuously. The long-term hourly runoff and soil-moisture can be simulated through the SURR model. Although evapotranspiration is not directly linked with flash flood forecasting, more realistic soil moisture estimation is possible by considering the evapotranspiration term. We added some references on the applicability of the SURR model (page 5, line 29~ page 6, line 1). "Bae and Lee (2011) showed that the SURR simulations are well fitted to observations, and Nash and Sutcliffe model efficiencies in the calibration and verification periods which are in the ranges of 0.81 to 0.95 and 0.70 to 0.94, respectively. Additionally, the behavior of soil moisture depending on the rainfall and the annual loadings of simulated hydrologic components are rational. From these results, an SURR model can be used for simulation of soil moisture."

4. I suggest changing the order of the methods and study area section. Providing first a general background on the region of interest and the data available before dealing with the method makes more logical sense to me.

-We agree, and we have changed the order of the methods and study area section. The revised section order is as follows. 2. Study Area and Datasets 3. Methods 3.1 QPC Computation 3.2 Flash Flood Guidance (FFG) 3.3 Receiver Operating Characteristics (ROC)

5. The results section starts off with more methods, uncomfortably emphasizing even more the logical disruption between the early sections of your manuscript. You offer some hydraulic geometry that you derive from a multiple regression model, in which the predictors are clearly correlated. This will need some more robust statistical treatment. Further down the section you mention that the predicted and observed timing of flash floods seem to be roughly similar. This is the first explicit mention of comparing predictions with observed data, and thus the motivation for using ROC curves, I presume. If so, please make sure that this core message comes across much earlier. Again, the time steps or measurement/simulation intervals here are critical. Please elaborate. I

am a bit suspicious about Fig. 10. Does basin area somehow play a role in estimating rainfall intensity in any of your models? Finally, your validation (section 4.4) needs to be more convincing. You mention that you tested your method on four observed flash floods between 2005 and 2009. How many cases did you use for training your classifier? Can you show some ROC curves (or other performance metrics) for the testing cases?

-Section 4.1 show the regional regression results for channel geometry. We added more analysis results and references because the main part of our paper is not this section (page 7, line 21∼24). "The derived regression equations are also shown in Table 3, and the determination coefficients of the regression equation were 0.76, 0.37 and 0.53 (Cho et al., 2011). The determination coefficient of hydraulic depth (H) is lower than the other variables. If additional data regarding river cross section are available, the regression equation will be improved."

-We claimed that FFG shows good performance in South Korea and that a ROC analysis can be applied by using these data instead of actual flash flood events (page 8, line 15-17). "As shown in Table 2 and Figure 7, the timing of the flash flood occurrence computed from the FFG model exhibited satisfactory agreement with those from the observed flash flood record."

-We performed a ROC analysis for all sub-basins. We estimated the virtual rainfall value associated with the peak ROC score. The results showed that virtual rainfall could be estimated as a function of the corresponding sub-basin area. These results show that the threshold for flash flooding can be classified by sub-basin area. -We added the method and assumption of validation (page 9, line 9-13). However, the P-A curve was trained by using FFG rather than actual flash flood events. The ROC score of a specific basin is determined as shown in Figure 8. "For the validation of the performance of the P-A curve, the quantitative flash flood criteria for actual flash flood events were applied. This experiment assumed the gauged mean areal precipitation as a prediction. The experiments were assessed whether the prediction exceeded

the quantitative flash flood criterion when an actual flash flood event occurred in the basins. If the prediction exceeded the quantitative flash flood criterion, a flash flood warning would be issued. According to the results, the flash flood occurrence was captured for 9 out of 12 events when the criteria were evaluated (Table 4)."

6. Your study could use a formal discussion section, in which you objectively discuss your methods in the light of their assumptions, limitations, and benefits (or advances) compared to previous work. Consider reflecting on how accurately SURR produces the necessary input data; how your classification would change for different time intervals; how your classification deals in general with rare events (for which ROC curves might not be the best of performance metrics); and what you consider as possible future improvements to your model.

-We added a discussion session, and this section was organized into two sub-sections (5.1Uncertainty of flash flood forecasting methods, 5.2 Utilization of a P-A curve for flash flood forecasting). The contents of 5.1 are different in this study from those in previous studies. The assumptions, limitations, and future work of this study are described (page 9 line 26~page10 line 17). The contents of 5.2 are reviews of flash flood forecasting systems used abroad, and the section discusses the usefulness of the P-A curve (page 10 line 18~page 11 line 3).

[revised manuscript text omitted]

7. Your conclusions mostly summarize your data. You report a high prediction potential, which is partly based on finding the optimal ROC scores in the first place, right? You state that 'The flash flood warning threshold can be best represented as a function of sub-basin area' (page 8/line 27). What does that mean and what is its practical relevance for warning? You may want to report statistical uncertainties for your generalized precipitation-area in this context.

-We added some sentences as described below "These results mean that the threshold for 1-hr flash flood prediction can be classified according to sub-basin area.". âŰžWe described the statistical uncertainties in the discussion section (page 10 line 10∼17).

8. The reference list appears a bit short. I imagine that other groups must be working on prediction of flash floods elsewhere.

-We performed that the literature review related to flash flood forecasting and added twenty papers as references (page 11, line 19-page 13, line 28).

9. Figures: #1 is OK, if you add some explanatory detail to the caption; please explain all abbreviations. #2 needs geographic coordinates and larger fonts. #3a and # 4a need units for 'sub-basin area'; are #3b and #4b really necessary? Histogram bins in #5b may be too wide: what is it that you wish to state here? #6 needs explanations of color codes. #7 needs larger fonts and explanation of abbreviations. #8: it is unclear what the minimum and maximum numbers refer to. #9: please explain orange shades. #10: please explain red and blue circles. Overall, you may want to use your captions for informing readers more about the contents and messages of your figures.

-We revised Figure 1 to provide more detail. -We added the unit of sub-basin area in Figure 3 and Figure 4. -Figure 3b and 4b are necessary for the understanding of sub-basin area distribution. -The interval of the histogram bins of Figure 5b was changed from 5 to 3. -We added some explanation of the color codes in Figure 6 -We changed the font size and explained the abbreviations of Figure 7 -The minimum value of Figure 8 was revised -We explained the orange shades of Figure 9 and the red and blue circles in Figure 10

10. Is Table 1 necessary?

-We think that this table is necessary because table 1 shows readers the concept of ROC analysis.

11. Please ask a native speaker to check your manuscript. I have noticed numerous formal and potentially ambiguous errors in the text, but these errors are too many to list in detail below. Therefore, I only give only a few examples in the line-specific suggestions below.

-We ordered English editing from AJE (American Journal Experts) and the manuscript has been revised by native English-speakers persons. We believe that these edits have

[Figure]

solved the English grammar problems and improved the readability of the text.

*Specific Suggestions

1. (1/8) Delete 'with short duration'. The term 'flash flood' implicates that.

-We deleted this term.

2. (1/9) 'required to cause minor flooding' - Why minor flooding? Please provide a brief definition of what you mean by 'minor' here.

-We revised that sentence as shown below (page 1, line 8~10). Generally, the threshold runoff of FFG is based on a 1~2 year return period flood. "Flash Flood Guidance (FFG), which was defined as the depth of rainfall of a given duration required to cause frequent flooding (1~2 year return period) at the outlet of a small stream basin"

3. (1/12) Please spell out 'ROC'. .

-We added the complete spelling of ROC.

4. (1/15) 'highly' should read 'more'?

-We revised 'highly' to 'more'

5. (1/16) 'obtained for rainfall rates of 42, 32 and 20 mm/h' - It is unclear why or how you picked those rates. Please explain.

-We revised the sentence as shown below, and we added a more detailed description (page 1, line 16~17). "For the brief description of the P-A curve, the generalized thresholds for flash flood warning can be suggested for rainfall rates of 42, 32 and 20 mm/h in sub-basins with areas of 22~40 $km^2$, 40~100 $km^2$ and >100 $km^2$, respectively.

6. (1/17) 'actual' means 'observed' or 'measured'? Please summarize briefly the results from your validation.

-We revised 'actual flash flood events' to 'observed flash flood events' (page 1, line 18).
-We added the validation results shown below (page 1, line 17~19). "The proposed
P-A curve was validated based on observed flash flood events in different sub-basins. Flash flood occurrences were captured for 9 out of 12 events."

7. (1/20) 'the short-duration flash flood frequently occurred' should read 'the flash floods occur frequently'.

-We revised 'the short-duration flash flood frequently occurred' to 'flash floods occur frequently'

8. (1/24) 'managing flash flood control' - What do you mean by that specifically?

-We revised that sentence as below (page 1, line 24~25). "It is difficult to monitor and forecast flash floods due to the unusually short response time for these natural disasters."

9. (1/25) 'the climate change has increased' could read 'climate change may have likely increased'.

-We revised 'the climate change has increased' to 'climate change likely increased' (page 1, line 25~26)

10. (1/27) What sort of 'technology' do you mean? Or did you mean 'methodology' instead?.

-We revised that sentence as shown below (page 1, line 27~27). "Therefore, reliable flash flood forecasting methods are necessary for flash flood response"

11. (1/28) 'For deciding flash flood occurrence,' - Unclear.

-We revised 'For deciding flash flood occurrence' to 'To judge flash flood occurrence' (page 1, line 28).

12. (2/1) 'flash flood vulnerability' - This refers to potential damage. Is that what you meant?

-We revised 'flash flood vulnerability' to 'flash flood vulnerability (possibility of flash

flood occurrence and degree of danger)' (page 2, line 1∼2).

13. (2/5) 'simulation to establish the observed frequency distribution' - Contradictory. Why simulate something to establish observations? Perhaps change the wording here?

-This phrasing is correct, because generally there are not enough runoff data in a small basin, so we need to simulate runoff data using a hydrological model. However, we revised this sentence as shown below to improve the sentence (page 2, line 4∼5). "However, this approach has some limitations for real-time flash flood forecasting because it requires long historical data and hydrological simulation to establish a flash flood modeling system."

14. (2/6) 'comparing forecast flow with flooding flow' - How about 'comparing forecast with observed flows'?

-We revised 'comparing forecast flow with flooding flow' to 'comparing forecasts with observed flows'

15. (2/7) Delete 'eminent'.

-We deleted 'eminent'.

16. (2/9) 'understood by the general public' - It may be useful to briefly explain the concept here.

-We revised the sentence as shown below (page 2, line 8∼10). "This method is commonly used for flash flood forecasting, as it is easily understood by the general public because it provides a qualitative criterion that can be used to intuitively determine whether a flash flood will occur."

17. (2/13-15) So what did those studies find out?

-We added their findings as below (page 2, line 17∼18). "They claimed that physically based methodologies are more appropriate for flash flood forecasting"

18. (2/13-15) 'the hourly maximum rainfall exceeded 50mm/hr and 60 mm/hr in 2006 and 2011' - Difficult to assess the relevance of these rates without any background information on rainfall characteristics in the region.

-We added the climatic characteristic of the basin in Section 2 Study area and Datasets (page 3, line 23-26) "The average annual precipitation was 1,390 mm, and the annual mean temperature was 11.5 °C over the 30 years of weather data from 1980 to 2009. More than 70% of the annual precipitation occurs during the flood season (June, July, August and September). The probability rainfalls for 1-hr at Seoul station are 52 mm/hr, 74 mm/hr, and 91 mm/hr for 3-year, 10-year, and 30-year return periods, respectively."

19. (2/23) Delete 'exquisitely'. Please also check grammar in this sentence. I think I know what you mean here, but you would be really well advised to seek the help of a native speaker for rephrasing many similar statements in your manuscript.

-We removed 'exquisitely' and revised the sentence as shown below (page 2, line 31∼32). "It is less important to estimate the soil moisture or runoff in the regions where flash floods occur frequently with short duration because the response time for a flash flood is limited"

20. (3/9) What are 'ROC scores'?

-We added some description of the ROC score (page 6, line 21∼25). "However, a ROC curve cannot be clearly indicated for objects that are more accurate than other objects. Wilk (2006) suggested an ROC Score which is the area of ROC curves. An ROC score can be calculated by using HR and FAR, as shown in Eq. (6)"

21. (3/15) 'method used to compute FFG is the opposite' - So what is the main output of FFG?

-We revised those sentences as shown below (page 4, line 21∼23). "The method used to compute FFG involves procedures opposite to those of a rainfall-runoff model. In other words, FFG is defined as the depth of rainfall over a given duration needed to

initiate flooding at the outlet of a small stream basin. It is generally estimated for 1-, 3-, and 6-hour durations."

22. (3/19) 'over a given duration tr required to' - Please use italics for all parameters that you introduce.

-We changed all parameters to italics (page 4, line 26~page 7, line 24).

23. (3/24) What is the unit of the 'unit hydrograph peak', if you use differing metric systems? Please attend to Equation 1: in my copy of the PDF it looks as if A is an exponent in the denominator.

- The units of q_pR are cfs/mi2/in. We added the units in the manuscript (page 5, line 1). -We revised Equation 1 as shown below (page 4, line 30).

24. (4/4) 'which represents current soil conditions' - What do you mean by 'current'? During or before the flash flood?

-We revised that sentence as shown below (page 5, line 14). "To derive the rainfall-runoff curve which represents soil conditions during flash flood event"

25. (4/9) 'this model uses estimates soil moisture' - Ambiguous. Does the model use estimates of soil moisture or does it estimate soil moisture itself? That is a big difference.

-The SURR model can estimate the soil moisture based on simulations of runoff and actual evapotranspiration. Thus, SURR can generate soil moisture, surface runoff, ground runoff, and actual evapotranspiration. SURR is described in detail in the manuscript (page 5, line 14~page 6, line 1).

26. (4/27) Please explain parameters in Equations 4 and 5.

-We added an explanation of the parameters in Equation 4 and 5 (page 6, line 7~9). "H and M represent hits and misses for predictions of when a flash flood will occur (OR > FFG). F and N represent false and negative hits for when a flash flood does not occur

(OR < FFG)."

27. (5/5) 'line segments that coincide with the left boundary and upper boundary of the ROC diagram' - You could simply say that, for a perfect prediction, the ROC curve has to pass through (0, 1) or the upper left point of the graph.

-The upper left point of the graph represents perfect prediction. We added this sentence to the manuscript (page 6, line 16∼17).

28. (5/7) 'ROC curves associated with real forecasts generally fall between these two extremes and plot above and to the left of the 45-degree diagonal' - Not sure what you mean by a 'real' forecast.

-We deleted this sentence because it is not necessary. However, we added some details about the ROC score.

29. (5/15) 'were delineated' - How did you delineate those basins? Their size spans three orders of magnitude, so what was the underlying rationale?

-We delineated the sub-basin using 30âĚź30 m DEM. The parameter of flow accumulation should be set to delineate the area of sub-basins in the range of 0∼100 km2.

30. (5/19) 'omitted from further analysis' - So you did not consider all basins with reservoirs further?

-Correct, we did not consider reservoir effects.

31. (5/21) 'filtering' - This means you had some preconception about basin area influencing flash-flood potential? It might be good to give more detail here.

-We revised those sentences as follows (page 3, line 16∼17). "Among the 660 sub-basins, we selected head water basins and mountainous basins and removed the artificial river basins. A total of 200 sub-basins were selected, as shown in Figure 3a."

32. (5/28) 'soil moisture conditions were estimated' - Please be more specific about

the spatial resolution, time intervals, and accuracies of those estimates.

-The SURR model was run for the hourly time series with a sub-basin scale. We now discuss the accuracy of the SURR model (page 5, line 29~page 6, line 1). "Bae and Lee (2011) showed that the SURR simulations are well fitted to observations, and Nash and Sutcliffe model efficiencies in the calibration and verification periods which are in the ranges of 0.81 to 0.95 and 0.70 to 0.94, respectively. Additionally, the behavior of soil moisture depending on the rainfall and the annual loadings of simulated hydrologic components are rational. From these results, an SURR model can be used for simulation of soil moisture."

33. (5/30) 'flood information was obtained through different sources, including print and electronic media' - How homogeneous and reliable is that information?

-Yes. The information about flash flood events is not homogeneous which is the source of flash flood forecasting uncertainty. We described this uncertainty in the discussion session.

34. (6/1) 'multiple flash flood events' - Perhaps this is something you may wish to elaborate on a bit more?

-We revised that sentence as follows (page 4, line 1~2). "In 2011, several flash flood events occurred with different areas and dates."

35. (6/15) 'were investigated and included in the regression equation' - Please describe this in more detail. You note that some of the predictors in your regression model are correlated, but you do not seem to do anything about this.

-We added more analysis of the results (page 7, line 21-24). "The derived regression equations are also shown in Table 3, and the determination coefficients of the regression equation were 0.76, 0.37 and 0.53 (Cho et al., 2011). The determination coefficient of hydraulic depth (H) is lower than the other variables. If additional data regarding river cross section are available, the regression equation will be improved."

36. (6/25) 'Threshold runoff values were computed' - How?

-We revised that sentence as follows (page 7, line 26~27). "The threshold runoff values were computed for effective rainfall durations of 1-hour in the 200 selected sub-basins by using the Manning equation and GIUH method, as mentioned in section 2.2."

37. (6/27) Can you measure runoff rate to one tenth of a mm/h?

-The threshold is calculated by using the Manning equation and GIUH; it is not measured. The units of threshold runoff is mm/hr or cm/hr.

38. (6/30) 'flooding season, i.e., July, August and September' - You could explain more about this flooding season in the study area descriptions; international readers might welcome this information.

-We added the climatic characteristic of the basin in Section 2 Study area and Datasets (page 3, line 23-26). "The average annual precipitation was 1,390 mm, and the annual mean temperature was 11.5 °C over the 30 years of weather data from 1980 to 2009. More than 70% of the annual precipitation occurs during the flood season (June, July, August and September). The probability rainfalls for 1-hr at Seoul station are 52 mm/hr, 74 mm/hr, and 91 mm/hr for 3-year, 10-year, and 30-year return periods, respectively."

39. (7/16) 'times of flash flood occurrence computed from the FFG model exhibited satisfactory agreement' - Is it the timing that you wish to classify correctly?

-We changed 'times' to 'timing'

40. (7/21) 'As expected, the minimum ROC score was 0.50' - You can sometimes get lower values than that.

-The range of ROC scores is 0.5 to 1.0 as shown in Figure 8 and Figure 9. We added more description of the ROC score (page 6, line 21-25). "However, a ROC curve cannot be clearly indicated for objects that are more accurate than other objects. Wilk (2006) suggested an ROC Score which is the area of ROC curves. An ROC score can be

calculated by using HR and FAR, as shown in Eq. (6)."

41. (8/12) 'estimated values of 1-hr QFFC' - Do you have measured values for a validation?

-We have the timing and locations of the flash flood. We can analyze the flash flood criteria when flash floods occur.

42. (8/29) 'optimum threshold for flash flood warning in a sub-basin' - Slight repetition.

-We replaced 'optimum threshold for flash flood warning in a sub-basin' with 'it'.

43. (9/4) 'which is divided with short and long-duration' - And how do set the threshold between 'short' and 'long'?

-We revised that sentence as follows (page 11, line 16~17). "Therefore, the development of a coupled flash flood forecasting system, which is divided into short (less than 1 hr) and long-duration (greater than 3 hrs) is necessary for managing flash flood efficiently."

Please also note the supplement to this comment:
https://www.nat-hazards-earth-syst-sci-discuss.net/nhess-2017-213/nhess-2017-213-AC1-supplement.pdf

———————————————

**Estimation of FFG**

**GIS Data**

√ DEM
√ Land Use
√ Slope
√ Soil_Depth
√ Soil_SAT
√ Soil_FC
√ Soil_WP

**Small Mountainous Basin**

**SURR**

**Threshold Runoff**

**Soil Water Deficit**

Meteoro-logical Data

Discharge Data

Rainfall Data

**FFG**

**MAP**

**Virtual rainfall**

√ 1-100mm
√ Increase 1mm

**ROC Analysis**

**Flash Flood Occurrence**
**FFG<MAP , FFG<VR**

**Compute HR, FAR**

**Compute ROC Score**

NO  VR=100mm ?  YES

**Quantitative Flash Flood Criteria**

**Extraction Max ROC Score**

**Virtual Rainfall of Max ROC Score = Flash Flood Criteria Rainfall Intensity**

**Derive Precipitation-Area Curve**

**Fig. 1.** Revised Figure 4

---

## Author Comment (AC2) · 30 Oct 2017

We thank you for your constructive review and comments. We have attached answers and the revised manuscript. Our responses were written after this symbol (-). We believe that the manuscript is in much better form now.

Referee's comments:

Important topic of high relevance. The paper is generally of a good structure and gives good insight in what has been done in the project. But not really a new approach, already been done in similar ways in other regions. Following open issues should be addressed in the publication: The cells of convective events are much smaller than the catchments described in the paper, so in the real world only part of the catchment

will be in the focus of the precipitation event. There is little info about what type of precipitation measurement has been used, are this ground measurements or radar or some combination of it? What is the resolution of the measurement? As the convective events are difficult to measure, the uncertainty applied by this also should be discussed. In the publication a lot of abbreviations are being used, this makes it hard to read, especially as they are not commonly used abbreviations, so better replace them by the full text. As the timely distribution of a rainstorm event also has high impact on the runoff, this should be tackled as well. Some more words should be spend on how the results can be used, in what extent are the usable for warning issues and how false warnings can be handled. Who is the planned end user of the thresholds?

-We revised the paragraph and added a description of the precipitation dataset as shown below (page 3, line 20~23). "Rainfall and soil moisture were the main datasets used to estimate Flash Flood Guidance. Rainfall data were obtained at 96 locations from the Ministry of Land, Infrastructure and Transport (MOLIT) and at 25 locations from the Korean Meteorological Administration (KMA). Rain gauges recorded data at 114 locations, and the resolution of each station was approximately 217 km2 (approximately 15*15 km)."

-We added a discussion section about the sources of uncertainty problems with this method (page 9 line 26~page10 line 16)

[revised manuscript text omitted]